# Increased Complement Activation and Decreased ADAMTS13 Activity Are Associated with Genetic Susceptibility in Patients with Preeclampsia/HELLP Syndrome Compared to Healthy Pregnancies: An Observational Case-Controlled Study

**DOI:** 10.3390/jpm14040387

**Published:** 2024-04-03

**Authors:** Theodora-Maria Venou, Evangelia Vetsiou, Christos Varelas, Angelos Daniilidis, Kyriakos Psarras, Evaggelia-Evdoxia Koravou, Maria Koutra, Tasoula Touloumenidou, Vasilis Tsolakidis, Apostolia Papalexandri, Fani Minti, Evdokia Mandala, Konstantinos Dinas, Efthymia Vlachaki, Eleni Gavriilaki

**Affiliations:** 1Hematological Laboratory, 2nd Department of Internal Medicine, Aristotle University of Thessaloniki, Hippokration General Hospital, 54642 Thessaloniki, Greeceevangelia.vetsiou1@nhs.net (E.V.); v_tsolakidis@hotmail.com (V.T.); 2Hematology Department, Papanicolaou General Hospital, 57010 Thessaloniki, Greece; cvarelas@auth.gr (C.V.); evakikor@gmail.com (E.-E.K.); tmdmgr@gmail.com (M.K.); tasoulatouloumenidou@gmail.com (T.T.); molbiol.gpapanikolaou@n3syzefxis.gov (A.P.); 31st Department of Obstetrics and Gynecology, Aristotle University of Thessaloniki, Papageorgiou General Hospital, 56429 Thessaloniki, Greece; angedan@auth.gr; 42nd Propedeutical Department of Surgery, Aristotle University of Thessaloniki, Hippokration General Hospital, 54642 Thessaloniki, Greece; psarrask@auth.gr; 5Department of Medicine, School of Health Sciences, Aristotle University of Thessaloniki, 54124 Thessaloniki, Greece; fani.minti@yahoo.com; 64th Department of Internal Medicine, Aristotle University of Thessaloniki, Hippokration General Hospital, 54642 Thessaloniki, Greece; emandala@auth.gr; 72nd Department of Obstetrics and Gynecology, Aristotle University of Thessaloniki, Hippokration General Hospital, 54642 Thessaloniki, Greece; 82nd Propedeutical Department of Internal Medicine, Aristotle University of Thessaloniki, Hippokration General Hospital, 54642 Thessaloniki, Greece; gavriiel@auth.gr

**Keywords:** preeclampsia, HELLP syndrome, pregnancy-associated liver disease, ADAMTS13, complement system, next-generation sequencing, C5b-9

## Abstract

Preeclampsia is a progressive multi-systemic disorder characterized by proteinuria, critical organ damage, and new-onset hypertension. It can be further complicated by HELLP syndrome (hemolysis, elevated liver enzymes, low platelets), resulting in critical liver or renal damage, disseminated coagulation, and grand mal seizures. This study aimed to examine the involvement of ADAMTS13, von Willebrand, and the complement system in the pathogenesis of preeclampsia/HELLP syndrome. We studied 30 Caucasian preeclamptic pregnant women and a control group of 15 healthy pregnancies. Genetic sequencing of ADAMTS13 and complement regulatory genes (MiniSeq System, Illumina) was performed. The modified Ham test was used to check for complement activation, ADAMTS13 activity, von Willebrand antigen (vWFAg) levels, and soluble C5b-9 levels were measured. Patients with preeclampsia had a decreased ADAMTS13 activity and increased C5b-9 levels. The vWFAg was significantly correlated with ADAMTS13 activity (r = 0.497, *p* = 0.003). Risk-factor variants were found in the genes of ADAMTS13, C3, thrombomodulin, CFB, CFH, MBL2, and, finally, MASP2. A portion of pregnant women with preeclampsia showed a decline in ADAMTS13 activity, correlated with vWFAg levels. These patients also exhibited an elevated complement activation and high-risk genetic variants in regulatory genes. Further research is needed to determine if these factors can serve as reliable biomarkers.

## 1. Introduction

Preeclampsia (PE) is a common pregnancy-specific disorder with high mortality and morbidity rates worldwide. As a clinical entity belonging to the broad group of hypertensive disorders of pregnancy, it is defined as new-onset hypertension combined with proteinuria or other final-stage organ damage after 20 weeks of gestation [1]. This health condition is also classified under the umbrella of pregnancy-related liver disorders and can be further complicated by HELLP syndrome (hemolysis, elevated liver enzymes, low platelets), resulting in critical liver or renal damage, disseminated coagulation, and the development of grand mal seizures (eclampsia) [2].

PE understanding has seen major advances over the last twenty years; however, the exact pathophysiologic mechanisms still need to be determined. Until today, placental maternal vascular and metabolic malfunctions have been identified as crucial factors in its pathogenesis. Several processes, such as mutations; endothelial and trophoblast dysfunction, including spiral artery remodeling, low oxygenation of the placenta, and oxidative stress; impairment of the immunotolerance at the maternal–fetal surface; and an imbalance among angiogenic and antiangiogenic factors, have been implicated as contributing factors [1,2]. Establishing new reliable biomarkers in diagnosing PE/HELLP syndrome is of unparalleled importance for the earliest possible diagnosis and treatment.

Among immune mechanisms, there is piling interest in the involvement of the complement system and its overactivation in PE. Under normal circumstances, the complement system, an innate defense mechanism, is suppressed during pregnancy, sustaining the maternal immunological tolerance and allowing the successful development of the fetus [3]. Sparse studies suggest that the complement pathways elude the regulating suppressing mechanisms and promote inflammation through augmented terminal activation, further damaging the utero–placenta unit and disrupting the placenta perfusion [3,4]. The terminal complement component C5b-9 (membrane attack complex) is commonly detected in urine and placental samples of preeclamptic women, while other earlier complement components are also identified in patient samples, indicating the participation of the complement cascade in the pathogenesis of PE [2,4,5].

Several studies have linked PE and HELLP to dysregulated hemostasis and thrombotic microangiopathies (TMAs), which are characterized by elevated levels of the von Willebrand factor antigen (vWFAg) and reduced plasma activity of ADAMTS13 (a disintegrin and metalloproteinase with a thrombospondin type 1 motif, member 13) [2,6]. Over the last decade, significant progress has been made in developing clinical and laboratory approaches to predict disease course and genotype–phenotype correlations. It has been reported that, in patients with TMAs, ADAMTS13 single nucleotide polymorphisms (SNPs) can lead to partial deficiency when a primary trigger is present or when a mutation in a complement gene is co-inherited, indicating that, except for the clinical characteristics, the implicated pathophysiological mechanisms are also similar [7].

The use of ADAMTS13 activity is important for patients diagnosed with preeclampsia for several reasons. Preeclampsia/HELLP syndrome shares clinical features with thrombotic microangiopathies (TMAs) such as thrombotic thrombocytopenic purpura (TTP) and hemolytic–uremic syndrome (HUS). The measurement of ADAMTS13 activity can help differentiate between these conditions. In TTP, there is a typically severe ADAMTS13 deficiency, leading to excessive von Willebrand factor activity and microthrombi formation, while in preeclampsia, ADAMTS13 activity is usually lower but within normal range. Nevertheless, investigating ADAMTS13 activity in preeclampsia can provide insights into the underlying pathophysiological mechanisms of the disorder. Dysregulation of the balance between vWF and ADAMTS13 has been implicated in endothelial dysfunction and thrombotic microangiopathy, which are central features of preeclampsia. In summary, measuring the ADAMTS13 activity in patients diagnosed with preeclampsia is important for accurate diagnosis, prognostication, guiding therapeutic decisions, and advancing our understanding of the pathophysiology of this complex disorder [1].

Our research is grounded in the hypothesis that additional factors beyond the established ones may contribute to the development of preeclampsia (PE) and HELLP syndrome. To explore this hypothesis, our study aims to investigate the potential involvement of ADAMTS13, von Willebrand factor, and the complement system in the pathogenesis of these conditions. Preeclampsia is a complex disorder with multifactorial origins and, while certain mechanisms have been identified, there remains a need to uncover novel pathways that could elucidate its etiology and improve clinical management. Therefore, our research endeavors to delve into the roles of ADAMTS13 and the complement system, both of which have been implicated in endothelial dysfunction and thrombotic microangiopathy, key features of PE and HELLP syndrome. Additionally, we seek to explore how variations in the genes associated with ADAMTS13 and the complement system might impact the occurrence and severity of these pregnancy complications. By investigating these potential contributors, we aim to expand our understanding of the pathophysiology of PE and HELLP syndrome, potentially paving the way for novel diagnostic and therapeutic strategies to improve maternal and fetal outcomes.

## 2. Materials and Methods

### 2.1. Study Population

We conducted an observational, case–control study examining pregnant women with early onset severe preeclampsia and healthy pregnant women with similar age and gestational weeks. All participants provided written consent after receiving comprehensive information about the study’s purpose and process. The study was conducted according to the guidelines of the Declaration of Helsinki and was approved by the Aristotle University of Thessaloniki Research Ethics Committee (protocol number: 22/5-2-2019).

Antenatal recruitment of pregnant women from the Department of Obstetrics and Gynecology of Hippokration General Hospital of Thessaloniki was conducted between July 2019 and March 2021. The diagnosis of preeclampsia was made in accordance with the American College of Obstetricians and Gynecologists criteria, which stipulates that a single incidence of systolic blood pressure (SBP) > 160; or diastolic blood pressure (DBP) > 110; or two instances of SBP > 140 or DBP > 90 with proteinuria or evidence of end-organ damage should be present [8]. Patients with pre-existing hypertensive conditions or a history of preeclampsia, HELLP syndrome, blood type O, antiphospholipid syndrome, or other autoimmune disorders and complement-mediated disease were excluded from the study. Medical records provided clinical variables such as systolic and diastolic blood pressure, as well as clinical laboratory parameters including random sample urine protein, serum creatinine, platelets, hemoglobin, glutamic-oxaloacetic transaminase (SGOT), glutamic-pyruvic transferase (SGPT), total bilirubin, and lactate dehydrogenase (LDH).

### 2.2. Laboratory Analysis

Peripheral blood and urine samples were collected on the day of enrolment. We performed genetic sequencing of the ADAMTS13 gene, as well as other complement regulatory genes (Complement factor H/CFH, CFH-related, CFI, CFB, CFD, C3, CD55, C5, MCP, thrombomodulin/THBD, MASP1, MASP2, MBL2, COLEC11, FCN1, and FCN3) with the use of Illumina MiniSeq sequencing System. DNA extraction from peripheral blood mononuclear was performed according to the Qiagen protocol.

The Modified Ham (mHam) test was utilized as an in vitro functional assay to evaluate complement activation. The mHam assay quantifies complement activation as a percentage of complement-mediated cell death, making it a functional assay. The testing process involves plating PIGA-null TF-1 cells into 96-well plates that contain gelatin veronal buffer with calcium and magnesium (GVB11; B102; Complement Technology Inc., Tyler, TX, USA). Subsequently, the cells are incubated at 37 °C for 45 min with diluted patient serum (1:5). After that, the cells are incubated at 37 °C for 2 h with diluted (1:10) WST-1 proliferation reagent (Roche, Basel, Switzerland). The absorbance is measured at 450 nm with a 630 nm filter in an ELISA plate reader (iMark Microplate Absorbance Reader, Biorad, Hercules, CA, USA). Using the sample’s absorbance and the heat-inactivated control’s absorbance, the percentage of non-viable cells is calculated to determine complement activation. The samples are tested in triplicates.

The ADAMTS13 activity and soluble C5b-9 levels were quantified using commercially available ELISA (enzyme-linked immunosorbent assay, iMark Microplate Absorbance Reader, Biorad) kits (Quidel, San Diego, CA, USA), following the manufacturer’s instructions. The measurement of vWFAg was carried out utilizing the immunoturbidimetric assay with the Siemens Healthineers BCS^®^ XP System. The ratios of vWFAg to ADAMTS13 activity (vWFAg/ADAMTS13) and platelets count to mean platelet volume (PLTs/MPV) were also calculated.

### 2.3. Bioinformatics and Statistical Analysis

The Statistical Package for Social Sciences (SPSS 27.0.1.0) statistical program for windows version 27 was used to perform statistical analysis in this study. Due to the small sample size and non-normal distribution of the investigated parameters, we utilized the non-parametric Mann–Whitney test to compare the two independent groups. Additionally, we performed Levene’s test to assess the equality of variances. The relationship between mHam test results and preeclampsia occurrence was discovered using the Chi-squared test. The level of statistical significance was set to 0.05. The median value and interquartile range of each parameter were calculated by group. Furthermore, a post hoc power analysis was conducted, which confirmed that the study population was adequate for a significance level of 5% and the desired power level of 80%.

To establish the clinical significance of the detected genetic variants, four bioinformatic tools (SIFT, PolyPhen-2, PROVEAN, and CADD) were used. It is worth mentioning that we considered every variant which was characterized as damaging by 2 or more bioinformatic tools as deleterious.

## 3. Results

### 3.1. Study Population

Table 1 presents the baseline characteristics of the two study groups. The patients’ group included 30 Caucasian preeclamptic pregnant women with a median age of 36 years [median age (IQR): 36 (10, 29–39)] and a median gestational age of 32 weeks. The control group, on the other hand, comprised 15 healthy pregnant women with a median age of 30 years [median age (IQR): 30 (11, 27–38) years] and a median gestational age of 30 weeks. Both groups shared similar baseline characteristics, and the control group had otherwise uncomplicated pregnancies.

The results of the study revealed a noteworthy disparity between the subjects with preeclampsia and the control group. Specifically, individuals with preeclampsia demonstrated a higher level of serum creatine and urine protein, with a statistically significant difference (*p* = 0.029 and *p* = 0.025, respectively). Additionally, a statistically significant difference was observed between the two groups concerning hemoglobin (*p* = 0.01), with the healthy control group displaying a lower median value. This notable difference could potentially be attributed to inadequate plasma expansion.

When it comes to liver function tests, it has been discovered that women who suffer from preeclampsia have significantly higher SGOT serum levels compared to healthy individuals in the control group (*p* = 0.003). However, there seems to be no significant difference in the levels of serum SGPT or total bilirubin between the two groups (*p* = 0.72 and *p* = 0.12, respectively). It is worth mentioning that none of the women in either group had elevated levels of serum total bilirubin. All participants in the healthy control group exhibited normal liver function test results. However, in the group of women with preeclampsia, three individuals showed elevated levels of SGOT, SGPT, or both, but none of them had a positive mHam test or increased C5b-9 levels.

Furthermore, it was interesting to find that ADAMTS13 activity was significantly lower in patients with preeclampsia [median (IQR, p25–p75): 70.1% (16.9, 65.3–82.2) vs. 89.5% (25.2, 71.8–97), *p* = 0.012], but remained within normal limits in both groups. Meanwhile, the levels of soluble C5b-9, as a biomarker of terminal complement activation, were increased in the group of patients with preeclampsia when compared to the healthy control [median (IQR, p25–p75): 311 ng/mL (386, 7–394) versus 35 ng/mL (5.2, 31–36.2)]. The difference was statistically significant between the two groups (*p* = 0.007). In addition, the modified HAM test was positive only in preeclamptic women (11/30, 36%) (*p* = 0.008). It is worth noting that all the patients in the preeclamptic group who tested positive for the mHam test also had elevated C5b-9 levels, indicating increased complement activation in this group of patients.

Moreover, the vWFAg was significantly correlated with ADAMTS13 activity (r = 0.497, *p* = 0.003). However, no statistically significant difference was observed between the two groups in terms of the vWFAg/ADAMTS13 ratio (*p* > 0.05). Furthermore, there was no statistically significant difference found between the two groups concerning the ratio of PLTs/MPV (*p*: 0.33 > 0.05). This ratio is expected to be lower in those with preeclampsia, as compared to healthy pregnant women due to a decline in PLTs count and an increase in MPV. Finally, no significant association was found between liver function tests (SGOT, SGPT, and total bilirubin) and C5b-9 levels or ADAMTS13 activity (*p* > 0.05).

### 3.2. Genetic Analysis

Conducting an initial evaluation to detect clinically significant variations by utilizing the ClinVar annotated data (analysis A), risk-factor variants were detected in the genes of ADAMTS13 (rs2301612), C3 (rs2230199), and Complement Factor H, CFH (rs800292). The finding of double heterozygosity in these genes was exclusive to preeclamptic women and, more specifically, to approximately half of them (16/30, 54%, *p* < 0.001). It is worth noting that, among the entire group of women with preeclampsia, only one individual exhibited indications of renal injury, as evidenced by a serum creatinine level of 1.51 g/dL and an estimated glomerular filtration rate (eGFR) of 43 mL/min per 1.73 m^2^, calculated using the MDRD equation: 175 × SerumCr-1.154 × age − 0.203 × 1.212 (if patient is black) × 0.742 (if female). This pregnant woman tested positive for the modified Ham test (C5b-9: 1232 ng/mL, ADAMTS13: 31%), and possessed double heterozygosity for high-risk genetic variants. Moreover, it should be mentioned that that there was an individual with preeclampsia who had a marked increase in liver enzyme levels (SGOT = 982 IU/L and SGPT = 1151 IU/L), but the mHAM test came back negative and normal C5b-9 serum levels were detected (C5b-9 = 49.2 ng/mL). In the preeclamptic pregnant woman with abnormal liver function tests, no high-risk genetic variants were identified.

Afterward, a further analysis was performed (analysis B) with the use of four bioinformatic tools and five deleterious genetic variants were identified in seven pregnant women suffering from preeclampsia. However, the same genetic variants were found only in three pregnant women in the healthy control group. The identified genetic variants are associated with the gene of ADAMTS13 (rs28503257 and rs28647808), thrombomodulin (rs1800579), CFB (rs12614), CFH (rs35274867), MBL2 (rs503073, rs1800450, and rs1800451), and finally MASP2 (rs12711521, rs139962539, and rs72550870). 

As per the study, patients who had clinically significant variants that were identified in both analysis A and analysis B (in the ADAMTS13, thrombomodulin, CFB, CFH, MBL2, MASP2, and C3 genes) showed a significant decrease in vWFAg levels compared to those who did not have such variants (177 vs. 295, *p* = 0.004). Conversely, patients with the rs1800450 variant in the lectin gene showed a significant increase in vWFAg levels (276 vs. 30, *p* = 0.038).

## 4. Discussion

The present non-interventional, case–control study supports the findings of other research groups. The study reveals a statistically significant reduction in ADAMTS13 activity and an elevation in soluble C5b-9 levels among a group of pregnant women who developed early onset severe preeclampsia, compared to a healthy control group. Furthermore, the investigation identified a number of genetic variants in the genes of ADAMTS13, C3, CFH, CFB, thrombomodulin, MBL2, and MASP2 that were significantly linked with the occurrence of preeclampsia, according to the results of analyses A and B. 

It has been a commonly held belief that the primary pathophysiological mechanism of preeclampsia is angiogenic imbalance, until recent times [3]. Endothelial cells and megakaryocytes are responsible for producing a complex glycoprotein called von Willebrand factor (vWF) in the form of preproprotein ultra large VWF (ULVWF). The ULVWF multimers are subsequently cleaved into less reactive vWF dimers and eliminated from circulation by ADAMTS13, which acts as its exclusive substrate. [6]. ADAMTS13’s involvement in the pathophysiology of PE/HELLP syndrome remains elusive. However, the common clinical presentation of TMAs, such as thrombotic thrombocytopenic purpura (TTP), in which ADAMTS13’s role is well understood [9,10], and PE/HELLP syndrome, suggests a linkage between the faulted expression of ADAMTS13 and the development of PE/HELLP syndrome [11].

An ADAMTS13 activity level of less than 10% is a strong indication of TTP diagnosis in relevant clinical contexts. However, several medical facilities lack the necessary equipment to perform the test on-site, necessitating the sending of samples to a reference laboratory, which further delays diagnosis. The differentiation between TTP and preeclampsia/HELLP syndrome is frequently based on coexisting clinical parameters. It is critical to distinguish between these two clinical entities, due to the contrasting management options. Delivery is the preferred treatment option for preeclampsia and the condition typically resolves within a few days after delivery. Nevertheless, delivery is not an appropriate treatment for TTP because symptoms continue to progress postpartum without plasmapheresis [12]. 

There is a question as to whether ADAMTS13 can be used as a reliable biomarker to diagnose preeclampsia/HELLP syndrome. In accordance with our findings, several studies found statistically significant decreases in ADAMTS13 activity in preeclamptic women’s serums over healthy controls [6,11,13,14,15,16,17]. A few articles, however, found no significant differences between the two groups of individuals, suggesting that ADAMTS13 activity is not associated with PE [18,19,20]. Consequently, besides the strong correlation between significantly low levels of ADAMTS13 activity (below 10–20%) and TTP, a more average level cannot rule out preeclampsia. However, the absence of ADAMTS13 activity cannot be equated with a diagnosis of preeclampsia [21]. 

Interestingly, in agreement with the results of our study, increasing evidence suggests that the terminal complement proteins C5a and C5b-9 are also involved in the pathogenesis of preeclampsia. Studies have suggested that PE is ultimately a complement-mediated disease, with the complement system being fundamental to its inflammatory process. The circulation of PE patients has been found to contain elevated amounts of complement components and their activation derivatives, such as C5a and C5b-9 complexes, even though the data are highly contradictory in this area [22]. It was also found that preeclamptic women’s trophoblasts accumulated more C5b-9 than non-preeclamptic women [23]. Moreover, in the study of Youssef et al., significantly higher levels of C5b-9 were detected in endothelial cells exposed to the plasma of pregnant women with severe preeclampsia at an early stage of pregnancy [5]. 

In a similar manner to our study, the involvement of the complement system in the hemolysis of severe PE was showcased in the study of Vaught et al., in which the complement activity was identified through a modified Ham test in ex vivo experiments [24]. It should be mentioned that the mHam test has been used as an indicator of complement system activation in other complement-mediated conditions, such as uremic hemolytic syndrome (aHUS) [25]. In another study, most patients with a diagnosis of HELLP had a positive mHam test (62%) compared to the healthy control group (16%), with a statistically significant difference between the two groups (*p* < 0.001). The results were similar in the group of patients with aHUS (88% versus 16%, *p* < 0.001) [26]. Therefore, even if a positive mHam test is not indicative of preeclampsia, the possibility of a complement system role in preeclampsia pathogenesis is highly plausible.

Moreover, the study of Burwick et al. demonstrated that there was no statistically significant difference between pregnant women with preeclampsia and pregnant women with chronic hypertension regarding plasma levels of C5b-9. This finding makes it unclear whether increased levels of plasma C5b-9 are specific to preeclampsia or merely suggest an increased risk of developing a number of clinical entities within the general category of hypertensive disorders of pregnancy. As part of the same study, C5b-9 excretion in urine was found to be a highly reliable biomarker for discriminating between severe preeclampsia and chronic hypertension in two different groups of pregnant women [27].

Liver disorders that may occur during pregnancy are unique and require special attention. There are several clinical conditions that can cause abnormal liver function tests during pregnancy, including hyperemesis gravidarum, intrahepatic cholestasis of pregnancy, acute fatty liver of pregnancy, and preeclampsia/HELLP syndrome [28]. It is important to keep in mind that pregnant women who have severe preeclampsia may encounter liver enzyme abnormalities, affecting up to 10% of cases. These abnormalities are usually identified by the presence of two to three times more alanine and aspartate aminotransferases than normal [29].

A study carried out by Valencia et al. showed significantly low plasma and high urine C5b-9 levels, which were correlated with end-organ damage in a group of pregnant women suffering from severe preeclampsia. It is possible that preeclampsia is caused by complement-mediated end-organ damage, as evidenced by the decreased plasma levels of complement system components combined with their increased excretion in urine. The same study found that hypertensive pregnant women without signs of organ dysfunction or other laboratory abnormalities did not exhibit a significant decrease in plasma C5b-9 levels [30]. We identified one woman who had signs of renal damage (increased creatinine and decreased eGFR) and high plasma levels of C5b-9 (C5b-9: 1232 ng/mL). In our study, no other individual in the preeclamptic group showed any signs of end-organ damage and this is possibly the reason why significantly increased levels of C5b-9 were measured in their plasma compared to the control group. 

Overall, according to the available research, complement control is disrupted in many women who develop preeclampsia and HELLP syndrome, and terminal complement activation is enhanced. These data imply that inhibiting the terminal complement pathway constitutes a promising field of investigation for new drug development in the future. In the literature, eculizumab, a C5 inhibitor, has been shown to be effective in treating severe preeclampsia, reducing the production of C5a. Pregnant women and fetuses are not believed to be at risk from eculizumab treatment [31,32].

Using next-generation sequencing (NGS), large genomic regions can be analyzed, allowing new possibilities to emerge. However, NGS has been slowly adopted for research on female reproductive health [33]. Based on the current knowledge of PE pathology, candidate gene studies have compared the genetic variation frequency among patients and controls to identify potential associations, such as the genes involved in endothelial function and blood pressure regulation (VEFGR-1, TGF-β, Eng, RAS, AGT, ACE, AGTR1, and eNOS), genes associated with lipid metabolism and oxidative stress (EPHX1, GST, NOX1, SOD2, APOE, LPL, and ROS), as well as thrombophilic genes (F5, F2, and MTHFR) [34]. In our study, risk-factor variants were found in ADAMTD13, C3, and Complement Factor H (CFH) genes. It is worth noting that the study conducted by von Krogh et al. did not find any correlation between the three investigated gene variants of ADAMTS13 and PE, including the rs28647808 variant, which our own study characterized as deleterious [20]. 

According to Lokki et al., the function of C3 can be affected due to variants in crucial domains of the C3 gene, simultaneously associated with the severity of the disease. Variant rs2230199 in C3 is known as the slow/fast mutation affecting the mobility of the protein during the electrophoresis process and was associated with the pathogenesis of PE [35]. On the other hand, other published studies found no correlation between this specific variant and preeclampsia [36,37]. Interestingly, according to Banadakoppa et al., the high-risk variant rs800292 in the CFH gene is associated with the activation of the alternative pathway in patients with preeclampsia, only in the coexistence of a variant in the CD46 gene [38].

A key strength of the present study is the use of standard laboratory techniques, such as ELISA, for measuring ADAMTS13 activity and C5b-9 activity, which could facilitate their establishment as biomarkers. A simple process, mHam, can also easily be incorporated into clinical practice in various complement-mediated diseases as an indicator of the activation of the complement system. 

The present study has some limitations worth mentioning. As a starting point, a larger sample size is necessary to increase the power of the study and reach valid conclusions. Collecting a larger sample can be challenging, especially for pregnant women with a preeclampsia diagnosis, since appropriate treatment might be required immediately. Due to the small sample size and the variability of illness in severe preeclamptic women, our conclusions are limited. The findings of this study cannot be generalized to women with mild preeclampsia or those with atypical disease. As a result of our findings, we can support the hypothesis that active disease in severe preeclampsia is typically associated with a dysregulation of the complement system. Nevertheless, in order to establish that complement indicators may indicate illness onset, additional research is required.

Finally, “Next Generation Sequencing” is a relatively new technology that has not been widely adopted, requiring specialized equipment and qualified personnel that are not always available. As a matter of fact, organizing a larger cohort is a difficult but significant undertaking. A further limitation of the present study is that only C5b-9 serum levels were measured. C5b-9 is part of the common pathway of complement activation. Therefore, we could not investigate which of the pathways (classical, lectin, or alternative) ultimately activates the complement system.

## 5. Conclusions

The results of our study indicate that a specific group of pregnant women suffering from preeclampsia demonstrated a decline in ADAMTS13 activity, which exhibited a positive correlation with vWFAg levels. These findings point towards potential endothelial damage and may warrant further investigation. Additionally, a subset of patients diagnosed with preeclampsia exhibit escalated complement activation, along with the presence of high-risk genetic variants in regulatory genes. Further research is required to determine whether these factors could potentially serve as dependable biomarkers for predicting the course of the disease, aiding in diagnosis, or even as promising therapeutic agents.

## Figures and Tables

**Table 1 jpm-14-00387-t001:** Baseline characteristics and laboratory tests of the study population [median (interquartile range—IQR)], level of statistical significance: 0.05 (Mann—Whitney test).

	Preeclamptic Group	Healthy Control Group	*p*
Age (years)	36 [10 (29–39)]	30 [11 (27–38)]	0.22
Gestational week	32 [3 (31–34)]	30 [9 (25–34)]	0.07
Hb (g/dL)	11.9 [2.1 (11.1–13.2)]	10.4 [2.4 (9.2–11.6)]	0.01
PLTs (/mm^3^)	219.000 [74.000 (187.500–261.500)]	253.000 [66.250 (206.500–272.750)]	0.60
SGOT (IU/L)	17.5 [10 (15–25)]	14 [4 (11–15)]	0.003
SGPT (IU/L)	16 [11 (12–23)]	11 [7 (8.25–15.25)]	0.72
Total bilirubin (mg/dL)	0.33 [0.18 (0.22–0.4)]	0.37 [0.12 (0.32–0.44)]	0.12
Serum Creatine (mg/dL)	0.71 [0.26 (0.62–0.88)]	0.67 [0.12 (0.6–0.72)]	0.029
Serum LDH (units/L)	191 [47 (175–222)]	190 [66 (160–226)]	0.43
Urine protein (mg)	129 [280 (18–298)]	0	0.025
Systolic Blood Pressure (mmHg)	165 [27 (150–177]	124 [13 (116–129)]	<0.001
Diastolic Blood Pressure (mmHg)	100 [20 (90–110)]	82 [11 (76–87)]	<0.001

Hb: Hemoglobin, PLTs: platelets, SGOT: serum glutamic-oxaloacetic transaminase, SGPT: serum-glutamic pyruvic transaminase, LDH: lactate dehydrogenase.

## Data Availability

The data presented in this study are not available due to ethical reasons.

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
