# Peer review of "Increased Complement Activation and Decreased ADAMTS13 Activity Are Associated with Genetic Susceptibility in Patients with Preeclampsia/HELLP Syndrome Compared to Healthy Pregnancies: An Observational Case-Controlled Study"

_jpm, 2024, doi:10.3390/jpm14040387_

Round 1

Reviewer 1 Report

Comments and Suggestions for Authors

In this study was indicated that

1.      ADAMTS13 activity was significantly lower in patients with preeclampsia compared to the healthycontrol70,1% (16,9, 65,3-82,2) vs. 89,5% (25,2, 71,8-97), p=0.012

2.      the vWFAg was significantly correlated with ADAMTS13 activity

3.      The finding of double heterozygosity in genes: DAMTS13 (rs2301612), C3

(rs2230199), and Complement Factor H, CFH (rs800292)was exclusive to preeclamptic women  (16/30, 54%, p<0.001)

Generally, the study is important and interesting work about potential factors that could contribute to the development of PE/HELLP syndrome with using NGS method which is undoubtedly an advantage of the work. This study has limitation including the small study population but analysis of individual cases is another advantage of the article.

I would like to suggest an improvement for the article:  

Line 131- 133:  ‘The ADAMTS13 activity and soluble C5b-9 were quantified using commercially available ELISA (enzyme-linked immunosorbent assay) kits (Quidel, San Diego, CA), following the manufacturer's instructions’ - ELISA plate reader should be added

Line: 161-162 only title should be in line 161, abbreviations should be under the table

Line 220-222: ‘the study revealed that patients with clinically significant variants exhibited a significant reduction in vWFAg levels as compared to those without such variants (177 vs 295, p=0.004)- In my opinion, it is difficult to understand if the authors analyse relationship between risk-factor variants which were found in the genes of ADAMTS13 (rs28503257), (rs28647808), thrombomodulin (rs1800579), CFB (rs12614), CFH (rs35274867), MBL2  (rs503073, rs1800450, rs1800451, MASP2 (rs12711521, rs139962539, rs72550870) – variants which were detected in patients with preeclampsia and in healthy control or in genes ADAMTS13 (rs2301612), C3 (rs2230199), Complement Factor H, CFH (rs800292)- variants which were detected in patients with preeclampsia  ?- The sentence should be corrected.

Line 229-230: in the discussion section: ,the investigation identified a number of genetic variants in the genes of ADAMTS13, C3, CFH, MBL2, and MASP2 that were significantly linked with the occurrence of preeclampsia’. In compared to the ‘results section’: The identified genetic variants are associated with the gene of ADAMTS13 (rs28503257  and rs28647808), thrombomodulin (rs1800579), CFB (rs12614), CFH (rs35274867), MBL2 (rs503073, rs1800450, rs1800451, and finally MASP2 (rs12711521, rs139962539,  rs72550870) and  line 199- 200: : Risk-factor variants were found in the genes of ADAMTS13 (rs2301612), C3  (rs2230199), and Complement Factor H, CFH (rs800292), I think that study should clearly indicate if the identified genetic variants are linked with the occurrence of preeclampsia, according to references or according to results of study. In my opinion, the sentence should be corrected (thrombomodulin and CFB were omitted ?)

Line 326- 329: according to the study, genetic variants  (rs387906343, rs142572218) were not found. The sentence should be corrected.

Author Response

Dear Reviewer 1,

We would like to express our sincere gratitude for the time and effort you have dedicated to reviewing our manuscript titled " Increased complement activation and decreased ADAMTS13 activity are associated with genetic susceptibility in patients with preeclampsia/HELLP syndrome compared to healthy pregnancies: an observational case-controlled study". Your valuable feedback and constructive criticism are greatly appreciated, and we believe they will significantly enhance the quality and impact of our work.

First, thank you for your thoughtful comments regarding the sample size of our study. We acknowledge that the sample size in our study is relatively small, and we understand the concern regarding its impact on the generalizability of our findings. It's important to highlight that our decision to work with a limited sample was primarily driven by the urgent nature of preeclampsia as a medical emergency. Preeclampsia presents significant challenges in terms of patient recruitment and retention due to its unpredictable onset and the critical need for prompt medical intervention. As a result, accessing a large and diverse sample size within a reasonable timeframe presents inherent difficulties.

We have carefully considered all the comments provided by the reviewer, and we are pleased to submit our response along with the revised version of the manuscript. Below, we address each of the reviewer's comments and outline the changes we have made in response:

  1. Line 131- 133: ‘The ADAMTS13 activity and soluble C5b-9 were quantified using commercially available ELISA (enzyme-linked immunosorbent assay) kits (Quidel, San Diego, CA), following the manufacturer's instructions’ - ELISA plate reader should be added

Response: We added the ELISA plate reader name (line 132-133)

  1. Line: 161-162 only title should be in line 161, abbreviations should be under the table

Response: Abbreviations were transferred under the table (lines 164-165).

  1. Line 220-222: ‘the study revealed that patients with clinically significant variants exhibited a significant reduction in vWFAg levels as compared to those without such variants (177 vs 295, p=0.004)- In my opinion, it is difficult to understand if the authors analyse relationship between risk-factor variants which were found in the genes of ADAMTS13 (rs28503257), (rs28647808), thrombomodulin (rs1800579), CFB (rs12614), CFH (rs35274867), MBL2  (rs503073, rs1800450, rs1800451, MASP2 (rs12711521, rs139962539, rs72550870) – variants which were detected in patients with preeclampsia and in healthy control or in genes ADAMTS13 (rs2301612), C3 (rs2230199), Complement Factor H, CFH (rs800292)- variants which were detected in patients with preeclampsia  ?- The sentence should be corrected.

Response: The sentence was corrected as follows: “As per the study, patients who had clinically significant variants that were identified in both analysis A and analysis B (in the ADAMTS13, thrombomodulin, CFB, CFH, MBL2, MASP2, and C3 genes) showed a significant decrease in vWFAg levels compared to those who did not have such variants (177 vs 295, p=0.004).” (lines 226- 229).

  1. Line 229-230: in the discussion section: the investigation identified a number of genetic variants in the genes of ADAMTS13, C3, CFH, MBL2, and MASP2 that were significantly linked with the occurrence of preeclampsia’. In compared to the ‘results section’: The identified genetic variants are associated with the gene of ADAMTS13 (rs28503257  and rs28647808), thrombomodulin (rs1800579), CFB (rs12614), CFH (rs35274867), MBL2 (rs503073, rs1800450, rs1800451, and finally MASP2 (rs12711521, rs139962539,  rs72550870) and  line 199- 200: : Risk-factor variants were found in the genes of ADAMTS13 (rs2301612), C3  (rs2230199), and Complement Factor H, CFH (rs800292), I think that study should clearly indicate if the identified genetic variants are linked with the occurrence of preeclampsia, according to references or according to results of study. In my opinion, the sentence should be corrected (thrombomodulin and CFB were omitted ?)

Response: We made sure to clarify that we conducted two separate analyses (lines 203-205 and 219-221). Additionally, we discussed thrombomodulin and CFB in the discussion section (line 238) and emphasized that the connection with preeclampsia was solely based on the outcomes of our study, rather than other references.

  1. Line 326- 329: according to the study, genetic variants (rs387906343, rs142572218) were not found. The sentence should be corrected.

Response: The sentence was corrected as follows in order to specifically refer to the variant rs28647808 which was also found in our study: “It is worth noting that the study conducted by von Krogh et al. did not find any correlation between the three gene variants of ADAMTS13 investigated and PE, including the rs28647808 variant, which our own study characterized as deleterious” (lines 336-338).

In addition to addressing the specific comments raised by the reviewer, we have also taken the opportunity to further refine certain sections of the manuscript to improve clarity, coherence, and overall readability. We are confident that the revisions we have made have strengthened the manuscript, and we believe it is now better positioned to contribute to the scientific community. We remain committed to ensuring the highest standards of quality and rigor in our research.

Once again, we would like to extend our gratitude to the reviewer for their invaluable feedback and guidance throughout this process. We look forward to hearing from you regarding the revised manuscript.

Thank you for your time and consideration.

Sincerely,

Venou Theodora Maria

Resident Doctor of Internal Medicin

Reviewer 2 Report

Comments and Suggestions for Authors

I had to peer review the manuscript “ Increased complement activation and decreased ADAMTS13 2 activity are associated with genetic susceptibility in patients 3 with preeclampsia/HELLP syndrome compared to healthy pregnancies: an observational case-controlled study”.

The topic is important and the manuscript well-written. However, in my opinion it is not clear from this manuscript why the use of ADAMTS13 activity is important for a patient diagnosed with preeclampsia.

Lines  87-90 :  The objective of this study is “The study aimed to examine the possible involvement of ADAMTS13,  von Willebrand, and the complement system in the pathogenesis of preeclampsia. Moreover, the study sought to explore how variations in the ADAMTS13 and complement system genes could impact the occurrence of preeclampsia/HELLP syndrome.” Can you be more precise about the objectives and the Results?

Comments on the Quality of English Language

English is fine. 

Author Response

Dear reviewer 2,

I am writing to express my sincere gratitude for your meticulous review of my manuscript titled "Increased complement activation and decreased ADAMTS13 activity are associated with genetic susceptibility in patients with preeclampsia/HELLP syndrome compared to healthy pregnancies: an observational case-controlled study," submitted for consideration in the “Journal of Personalized Medicine”.

Based on your constructive comments, we have made significant enhancements to the manuscript, particularly in clarifying the main objectives of the study. We emphasized the primary objective of investigating the possible pathogenetic role of ADAMTS13, as well as the complement system, in the pathogenesis of preeclampsia/HELLP syndrome. Additionally, we aimed to determine whether the measurement of ADAMTS13 can serve as a valuable biomarker for the differential diagnosis from thrombotic microangiopathies. Accurate differentiation between these conditions is crucial for appropriate clinical management, as treatment strategies differ significantly (lines 99-113).

We would also like to note that the use of ADAMTS13 activity is important for patients diagnosed with preeclampsia due to several reasons. Preeclampsia/HELLP syndrome shares clinical features with thrombotic microangiopathies (TMAs) such as thrombotic thrombocytopenic purpura (TTP) and hemolytic-uremic syndrome (HUS). Measurement of ADAMTS13 activity can help differentiate between these conditions. In TTP, there is typically severe ADAMTS13 deficiency leading to excessive von Willebrand factor activity and microthrombi formation, while in preeclampsia, ADAMTS13 activity is usually lower but within normal range. Nevertheless, investigating ADAMTS13 activity in preeclampsia can provide insights into the underlying pathophysiological mechanisms of the disorder. Dysregulation of the balance between vWF and ADAMTS13 has been implicated in endothelial dysfunction and thrombotic microangiopathy, which are central features of preeclampsia. In summary, measuring ADAMTS13 activity in patients diagnosed with preeclampsia is important for accurate diagnosis, prognostication, guiding therapeutic decisions, and advancing our understanding of the pathophysiology of this complex disorder (lines 85-98).

Moreover, I am delighted to inform you that we have made significant improvements to the results section of the manuscript. These enhancements have allowed for a more comprehensive and nuanced presentation of our findings, thereby strengthening the overall impact of the study.

Your expertise in the field has played a pivotal role in guiding us toward these improvements, and I am confident that the revised manuscript now more effectively contributes to the scientific discourse on the pathogenesis and diagnosis of preeclampsia/HELLP syndrome.

Best regards,

Venou Theodora Maria

Round 2

Reviewer 2 Report

Comments and Suggestions for Authors

The manuscript was improved. 

Comments on the Quality of English Language

English are fine, minor editing  is requiring.